# Non-medical determinants of perinatal health: protocol for a systematic review with meta-analysis

Leonie A Daalderop,[1,2] Marjolein W de Groot,[1] Lindsey van der Meer,[1] Eric A P Steegers,[1] Loes C M Bertens[1]

¹Department of Obstetrics and Gynaecology, Erasmus MC, Rotterdam, The Netherlands
²Dutch Research Institute for Transitions, Erasmus University Rotterdam, Rotterdam, The Netherlands

**Correspondence to**
Leonie A Daalderop;
l.daalderop@erasmusmc.nl

## ABSTRACT

**Introduction** Research focusing on the associations between non-medical determinants and unfavourable perinatal health outcomes is increasing. Despite increasing knowledge on this theme, it still remains unclear to what extent social, environmental and lifestyle factors contribute to these unfavourable outcomes. Therefore, we aim to provide a systematic review, preferably with meta-analysis, in order to provide insight into the associations between non-medical determinants and perinatal mortality, preterm birth and being small for gestational age (SGA).

**Methods and analysis** Observational studies performed in European countries studying the associations between non-medical determinants and unfavourable perinatal health outcomes will be included. Primary outcomes of interest are perinatal mortality, preterm birth and SGA. To retrieve potential eligible articles, a systematic literature search was performed in the following online databases on 5 October 2018: MEDLINE, Embase, Web of Science, Cochrane and Google Scholar. Additionally, a reference list check and citation search will be performed. Data of the included articles will be extracted using a standardised and piloted data extraction form. Risk of bias will be assessed using the Newcastle-Ottawa Scale. The study selection and data extraction process will be performed by two reviewers independently. Disagreements will be resolved through discussion with a third reviewer. The pooled effects will be calculated separately for each association found between one of the outcome measures and the non-medical determinants using a random effects model. Heterogeneity of the studies will be assessed using the I² statistic.

**Ethics and dissemination** No ethical approval is necessary for a systematic review with meta-analysis. The findings will be published in a peer-reviewed journal.

**PROSPERO registration number** CRD42018056105.

## Strengths and limitations of this study

▶ This systematic review with meta-analysis will provide a comprehensive overview of the existing literature on the associations between non-medical determinants and perinatal mortality, preterm birth and being small for gestational age.

▶ The results of the meta-analysis can highlight the relevance of the studied non-medical determinants, as well as the need to develop preventive strategies aimed at diminishing unfavourable perinatal health outcomes.

▶ This systematic review focuses on studies performed in European countries with more or less comparable healthcare systems, which limits the generalisability of our findings to countries outside Europe with different healthcare systems and/or a different welfare state.

occurrence of unfavourable perinatal health outcomes (ie, perinatal mortality, preterm birth and SGA).[4–9] These factors encompass, for instance, educational level, marital status, household income and housing conditions. The negative impact of non-medical determinants on perinatal health outcomes extends into adulthood, with long-term health consequences for the affected children and therewith major implications for public health.[10]

During the past decade, many studies have tried to identify the effects of non-medical determinants on unfavourable perinatal health outcomes.[4 6–9 11–19] Several studies reported non-medical determinants as risk factors for unfavourable perinatal health outcomes.[4 6 7 11–14] Additionally, studies indicate that the negative impact of non-medical determinants is often the result of clustering of two or more risk factors.[8 9] However, many other studies do not show associations between non-medical determinants and unfavourable perinatal health outcomes.[15–19] Furthermore, research on this topic often focuses on the associations between a limited number of non-medical determinants and a

## INTRODUCTION

Preterm birth and being small for gestational age (SGA) affect 11%–27% of all live births worldwide and are the leading causes of perinatal mortality.[1–3] Besides the negative impact of well-known medical and obstetric risk factors, risk accumulation of non-medical determinants related to a person's social status and environmental surroundings can explain the additional variation in the

single outcome measure, such as preterm delivery, low birth weight or SGA. Non-medical determinants, which are often related to a person's social status, are highly associated. Therefore, it is important to investigate these different non-medical determinants together, as well as the way they influence each other. This, together with the global burden of both non-medical determinants and unfavourable perinatal health outcomes, a comprehensive overview of relevant non-medical determinants and the way they affect perinatal health outcomes, is necessary to deliver the best possible care for vulnerable pregnant women.[3 20 21]

Therefore, we aim to provide a systematic review, preferably with meta-analysis, in order to investigate the impact of a broad range of non-medical determinants on unfavourable perinatal health outcomes. This can help scientists as well as healthcare providers to thoroughly understand the role of non-medical determinants with regard to unfavourable perinatal health outcomes.

## METHODS AND ANALYSIS

The Preferred Reporting Items for Systematic Reviews and Meta-analyses Protocols (PRISMA-P) guidelines were used to guide the reporting of this systematic review protocol (online supplementary file 1).[22] The initial search was performed on 5 October 2018, and the study was registered with the PROSPERO prospective register of systematic reviews on 12 November 2018. The expected finalisation of the systematic review with meta-analysis is end 2019.

### Eligibility criteria

Studies will be selected according to the criteria outlined further.

### Study design and participants

We will include all observational studies (ie, cohort, case–control and cross-sectional studies) reporting on pregnant women that study the associations between exposure to non-medical determinants during pregnancy and unfavourable perinatal health outcomes (ie, fetal complications during pregnancy, labour and/or delivery). Exposures have to be measured at the individual level. Studies performed in European countries are eligible for inclusion, as these are countries with a more homogeneous population and, more or less, comparable healthcare systems. Experimental studies, case reports, editorials, commentary and clinical guidelines will be excluded. Additionally, systematic reviews and meta-analyses will be excluded; however, their reference lists will be screened for potential eligible articles. Only articles written in English will be included.

### Determinants of interest

This systematic review will focus on non-medical determinants that may influence perinatal health outcomes.

Non-medical determinants that will be taken into account were prespecified. These non-medical determinants are
► Socioeconomic status.
► Marital status.
► Domestic violence.
► Employment status.
► Educational level.

After title and abstract screening, three other non-medical determinants were identified, which were not specified beforehand. These non-medical determinants were often investigated in relation to unfavourable perinatal health outcomes, and therefore we decided to add these determinants to our systematic review:
► Housing (eg, place of residence and housing conditions).
► Income.
► Paternal determinants (eg, non-medical determinants related to the father).

### Outcomes

The primary outcomes are
► Perinatal mortality: death occurring between 22 weeks of gestational age and 7 days after birth.
► Preterm birth: delivery of a live-born baby before 37 completed weeks of gestation.
► SGA: birth weight below the 10th centile adjusted for ethnicity, parity, gestational age and gender.
Secondary outcomes are
► Neonatal mortality: death of a baby occurring within the first 28 days of life.
► Low birth weight: birth weight below 2500 g.

It is possible that studies have used outcome definitions that differ from the ones defined earlier. Studies are in that case still eligible to be included into the systematic review. For the meta-analyses of the effects per outcome, only the effects from studies with a sufficient amount of overlap in the used definitions will be pooled.

### Information sources and search strategy

The following electronic online databases were initially searched on 5 October 2018: MEDLINE, Embase, Web of Science, Cochrane and Google Scholar. The electronic search yielded 4980 references. The search strategy will be supplemented by screening reference lists and performing a citation search of the included articles. Also, the search will be updated after finalisation of the data extraction process. A search strategy was developed for Embase (table 1), using Emtree terms and free text terms related to non-medical determinants, perinatal health outcomes, pregnant women and the study design. The search strategy was amended for use in other databases. No limitations will be applied regarding publication date.

### Study selection

All articles identified with the database searches will be uploaded or manually entered into EndNote X8.2 reference management software (Thomson Reuters, New York

**Table 1** Search strategy in Embase

| Element | Search items |
| --- | --- |
| Non-medical determinants | ('social determinants of health'/exp OR 'marriage'/de OR divorce/de OR 'female by marital status'/exp OR 'educational status'/exp OR 'unemployment'/de OR 'domestic violence'/de OR 'battered woman'/de OR 'family violence'/de OR 'partner violence'/exp OR 'family conflict'/exp OR 'social status'/de OR (((social* OR socioeconom* OR socio-econom* OR Socio-cultur*) NEAR/6 (determinant* OR status OR differen* OR correlat* OR predict* OR disadvantage* OR indicator* OR risk-factor* OR level* OR variable* OR vulnerab*)) OR (individual* NEAR/6 socioeconom* NEAR/6 (factor* OR determinant*)) OR divorce* OR low-school* OR (education* NEAR/3 (status* OR level OR matern* OR mother*)) OR unemploy* OR unmarried OR singlehood OR single-mother* OR ((domestic* OR famil* OR partner* OR spous* OR marital OR marriage OR household* OR house-hold*) NEAR/3 (status* OR violen* OR conflict* OR problem*))):ab,ti) |
| Perinatal outcomes | ('fetus mortality'/exp OR 'fetus death'/de OR 'infant mortality'/de OR 'perinatal mortality'/exp OR 'low birth weight'/exp OR 'intrauterine growth retardation'/de OR 'immature and premature labour'/de OR (((fetus OR fetal OR fetus OR foetal OR perinatal* OR newborn* OR new-born* OR neonat* OR infant* OR antepart* OR antenatal* OR perinatal*) NEAR/3 (mortalit* OR surviv* OR fatal* OR death*)) OR (small NEAR/3 (date OR gestation*)) OR sga OR lbw OR vlbw OR elbw OR (low NEAR/3 (birth-weight* OR birth weight*)) OR ((intrauterin* OR intra-uterin* OR fetus OR fetal OR fetus OR foetal) NEAR/6 (retard* OR restrict*)) OR IUGR OR preterm* OR pre-term* OR ((prematur* OR immature OR dysmatur*) NEAR/3 (birth* OR childbirth OR born OR neonat* OR infan* OR labour OR labour))):ab,ti) |
| Population | ('pregnancy'/exp OR 'pregnant woman'/exp OR 'prenatal period'/exp OR 'mother'/de OR 'expectant mother'/de OR 'maternal behaviour'/de OR (pregnan* OR prenatal* OR antenatal* OR mother* OR maternal*):ab,ti) |
| Study design | ('cohort analysis'/exp OR 'longitudinal study'/de OR 'retrospective study'/de OR 'prospective study'/de OR register/de OR 'factual database'/de OR (cohort* OR longitudinal* OR retrospectiv* OR prospectiv* OR register OR database*):ab,ti) NOT ((Conference Abstract)/lim OR (Letter)/lim OR (Note)/lim OR (Editorial)/lim) AND (english)/lim |

City, NY, USA). First, duplicates will be removed automatically and manually. Two reviewers (alternately LAD, MWdG or LvdM) will independently screen titles and abstracts of the retrieved articles to assess eligibility for inclusion. After the initial selection, full texts of potentially eligible articles will be obtained and screened to assess eligibility for final inclusion. Disagreements will be resolved through consultation of a third reviewer (LCMB).

### Data extraction
Two reviewers (alternately LAD, MWdG or LvdM) will independently extract relevant data from the included articles using a predefined data extraction form. This data extraction form will be piloted using the first five eligible articles and modified if required. The following data will be extracted from each article: author, publication year, study design, study setting, study period, number of included participants, study population characteristics, definition and measurements of the studied non-medical determinant(s), crude and adjusted effect estimates (ie, odds ratio (OR) or relative risk (RR)), covariates used for adjustment, reported limitations and key conclusions. If relevant information cannot be retrieved from the published articles, the authors of the manuscript will be contacted to request additional data.

### Risk of bias assessment
To determine the risk of bias of the included studies, the Newcastle-Ottawa Scale (NOS) will be used.[23] The

NOS is a validated tool for assessing the risk of bias among studies with an observational design. This tool is a commonly used tool for the assessment of risk of bias and is considered the most practical tool in use compared with other tools.[24 25] Two reviewers (alternately LAD, MWdG or LvdM) will independently assess the risk of bias of the included articles. Disagreements will be resolved through discussion with a third reviewer (LCMB).

### Data analysis
ORs and 95% CIs will be extracted from the included articles to express the effect of the associations between the studied non-medical determinants and perinatal mortality, preterm birth and SGA. Data presented in other effect measures (eg, RRs and beta coefficients) will be converted into ORs when possible. Publication bias will be evaluated using funnel plots and corresponding Begg and Egger tests. The degree of heterogeneity among the included articles will be examined using the $I^2$ statistic. The pooled effects will be calculated when there is a sufficient overlap among the used definitions among the included studies. When a meta-analysis can be performed, the pooled effect estimates will be calculated separately for each association found between one of the outcome measures and the studied non-medical determinants using a random effects model. The meta-analysis will be performed with both the unadjusted and adjusted estimates due to the expected heterogeneity in

the degree of adjustment. All analyses will be performed using R package 'metafor'.

## Patient and public involvement

Patients and/or the public will not be involved in this study.

## Ethics and dissemination

No formal ethical assessment or informed consent is required for the purpose of this study. In accordance with the PRISMA-P guidelines, the study is registered with PROSPERO (12 November 2018).[22 26] The findings of this study will be summarised in a manuscript, which will be submitted for publication in a peer-reviewed scientific journal.

**Contributors** EAPS, LCMB and MWdG conceived this systematic review with meta-analysis. LAD and MWdG developed and drafted the protocol. MWdG registered the protocol in PROSPERO. EAPS, LCMB and LvdM reviewed the protocol and provided extensive feedback. All authors read and approved the final manuscript.

**Funding** The authors have not declared a specific grant for this research from any funding agency in the public, commercial or not-for-profit sectors.

**Competing interests** None declared.

**Patient consent for publication** Not required.

**Provenance and peer review** Not commissioned; externally peer reviewed.

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
