## [Reviewer comments · BMJ Open]

ARTICLE DETAILS

TITLE (PROVISIONAL)	Non-medical determinants of perinatal health: protocol for a systematic review with meta-analysis
AUTHORS	Daalderop, Leonie; de Groot, Marjolein; van der Meer, Lindsey; Steegers, Eric; Bertens, Loes

VERSION 1 – REVIEW

REVIEWER	Dr Deborah Randall University of Sydney, Australia
REVIEW RETURNED	27-May-2019

GENERAL COMMENTS	This protocol is for a systematic review of non-medical determinants and their association with perinatal outcomes such as small for gestational age, perinatal death and preterm birth. Comments: 1. How will the authors deal with differences in registration of stillbirths by different European countries? E.g. Mohangoo AD, Buitendijk SE, Szamotulska K, et al. Gestational age patterns of fetal and neonatal mortality in Europe: results from the Euro-Peristat project. PLoS ONE 2011;6:e247272. How will the authors handle congenital anomalies? Will studies including congenital anomalies be included?3. Most of the specific non-medical determinants of interest like socioeconomic status, employment status, and education level will be highly associated. Will adjusted or unadjusted estimates be included in the meta-analysis?4. Why are ORs being used rather than RRs? Is the PAF formula appropriate for ORs?5. Also, according to Rockhill et al, the nominated PAF formula is not appropriate for use when there are confounders. Has this been considered by the authors? Rockhill B, Newman B, Weinberg C (1998) Use and misuse of population attributable fractions. Am J Public Health 88: 15–19.6. For the search terms, what is “female by marital status”? What is “low-school”?7. I cannot see search terms for “insurance status”, “neighbourhood deprivation”, or “housing”. Suggest including search terms along the lines of: - private health insurance, private medical insurance
--

	 - neighbourhood, neighborhood, urban, city, town, disadvantage, deprivation, poverty, disadvantaged community - housing, homeless, inadequate housing, insecure housing, precarious housing, unstable housing, public housing Further search terms to consider might be:  - disparity - occupation - employment - income - inequality - welfare - Carstairs /Townsend score (any other country-specific deprivation scores?) Minor comments: Page 3, line 43: “which limits the generalisability of our findings to countries with different health care systems” – slightly confusing sentence. Sound like the study is only generalisable to countries with different health systems, rather than the other way around. Suggest rewording. Page 4, line 6: Should this be “leading causes” rather than “leading cause”?
--	---

REVIEWER	Beogo Idrissa Canada
REVIEW RETURNED	01-Jul-2019

GENERAL COMMENTS	Major comments  1) From lay readers perspective, it is absolutely indicated to define non-medical determinants. In p417, the authors mention this (e.g. social, environmental, and lifestyle factors). It is not sufficient to light readers. 2) I was expecting the author to show up preliminary results in terms of the number of references from included databases. 3) Expected results are welcome under the form of discussion and or conclusion Minors comments P4116-7: Is there a reference to associate to this assertion? P4121-22: This is per see a limitation for any study. Could the author lay out more clearly the underlining idea? P5154: Although MEDLINE and PubMed are similar, the latter being the public interface, always provide more output as very recent article are posted. To stick on what you did -or plan to do- it is recommended to opt for on of them. P6124: It is weird the presence of this key word in the equation. I need the author to convince me as it seems contradictory ((mortalit* OR fatal* OR death*) OR surviv*).
--

VERSION 1 – AUTHOR RESPONSE

Reviewer: 1
Comments:

1. How will the authors deal with differences in registration of stillbirths by different European countries? E.g. Mohangoo AD, Buitendijk SE, Szamotulska K, et al. Gestational age patterns of fetal and neonatal mortality in Europe: results from the Euro-Peristat project. PLoS ONE 2011;6:e24727 Perinatal mortality, defined as any death occurring between 22 weeks of gestation and 7 days after birth, is one of our primary outcome measures and stillbirths are part of this outcome. It is known that European countries differ in their definition and registration of stillbirths. Potentially, comparable differences in other outcome measure definitions can be encountered. The definitions of stillbirth of the included articles will be systematically reported, as well as their reported findings, as part of the systematic review. For the meta-analysis, only the effects from studies using definitions that sufficiently overlap will be pooled. We have added a paragraph to the description of the outcomes, clarifying how we deal with differences in used definitions between studies. (page 5; "It is possible ... will be pooled." and page 6; "The pooled effects ... random effects model.").

2. How will the authors handle congenital anomalies? Will studies including congenital anomalies be included?

All eligible studies will be included, and the presence or absence of congenital anomalies is not an exclusion criterion of this systematic review with meta-analysis, with the exception of studies solely focussing on births with congenital anomalies. However, we will report whether studies adjusted for the presence of congenital anomalies.

3. Most of the specific non-medical determinants of interest like socioeconomic status, employment status, and education level will be highly associated. Will adjusted or unadjusted estimates be included in the meta-analysis?

We are aware of the fact that the non-medical determinants related to a person's socioeconomic status will be highly associated. We will therefore perform the meta-analysis with both unadjusted and adjusted estimates in an effort to try and capture this. (page 6; "The meta-analysis will ... degree of adjustment.").

4. Why are ORs being used rather than RRs? Is the PAF formula appropriate for ORs?

We include cohort, cross-sectional and case-control studies in our systematic review. While both measures can be calculated from cohort and cross-sectional studies, in case-control only odds ratios can be calculated. Therefore, only odds ratios will be calculated for all included studies.

5. Also, according to Rockhill et al, the nominated PAF formula is not appropriate for use when there are confounders. Has this been considered by the authors? Rockhill B, Newman B, Weinberg C (1998) Use and misuse of population attributable fractions. Am J Public Health 88: 15–19.

We appreciate your comment on the use of population attributable fraction (PAF). We carefully reviewed the available literature on the use of PAFs and based on that we have decided that we will no longer calculate the PAFs. We cannot assume that the estimated effects which we will use to calculate the PAF are adjusted for all confounders. On top of that, the risk factors that we investigate such as socioeconomic status, educational level, employment status and marital status are surrogates for more proximate exposures¹ and therefore, not really suitable for the calculation of PAFs. We deleted the text about the PAF in the manuscript. (page 2; "In addition to ... the European context." and page 6; "In addition to ... the non-exposed group.").

6. For the search terms, what is "female by marital status"? What is "low-school"?

The search term "female by marital status" is an Emtree term in Embase and includes the search terms divorced women, married women, single women and widow. The search term low-school* will retrieve phrases such as "low school achievement", "low school performance", "low school grades" ,

“low school marks” , “low school level” and “low schooling” among others. These are relevant for lower educational status.

7. I cannot see search terms for “insurance status”, “neighbourhood deprivation”, or “housing”. Suggest including search terms along the lines of:

- i. private health insurance, private medical insurance
- ii. neighbourhood, neighborhood, urban, city, town, disadvantage, deprivation, poverty, disadvantaged community
- iii. housing, homeless, inadequate housing, insecure housing, precarious housing, unstable housing, public housing
- iv. Further search terms to consider might be:
 - a. Disparity
 - b. Occupation
 - c. Employment
 - d. Income
 - e. Inequality
 - f. Welfare
 - g. Carstairs/Townsend score (any other country-specific deprivation scores?)

We pre-specified a number of non-medical determinants which were incorporated into the systematic search strategy. During the title and abstract screening another three non-medical determinants were identified which were not pre-specified. We decided to add these non-medical determinants to our systematic review since they were often investigated in relation to unfavourable perinatal health outcomes. We amended the text underneath the subheading ‘determinants of interest’ to clarify this. In response to your comment, we have critically reviewed the included non-medical determinants and we have decided to remove neighbourhood deprivation from this list. Neighbourhood deprivation was not included into the search terms because a recent systematic review with meta-analysis from Vos et al.² already investigated the association between neighbourhood deprivation and unfavourable perinatal health outcomes. The list of determinants of interest is now adjusted. (page 4; “This systematic review ... to the father.”).

Minor comments:

Page 3, line 43: “which limits the generalisability of our findings to countries with different health care systems” – slightly confusing sentence. Sound like the study is only generalisable to countries with different health systems, rather than the other way around. Suggest rewording.

We have changed this sentence. (page 2; “This systematic review ... different welfare state.”).

Page 4, line 6: Should this be “leading causes” rather than “leading cause”?

We have changed this. (page 3; “Preterm birth and ... of perinatal mortality.”).

Reviewer: 2

Please leave your comments for the authors below Major comments

1. From lay readers perspective, it is absolutely indicated to define non-medical determinants. In p417, the authors mention this (e.g. social, environmental, and lifestyle factors). It is not sufficient to light readers.

We appreciate your comment to clarify the definition of non-medical determinants. We changed the sentence used to define non-medical determinants. (page 3; “Besides the negative ... and housing conditions.”).

2. I was expecting the author to show up preliminary results in terms of the number of references from included databases. Expected results are welcome under the form of discussion and or conclusion

We appreciate the reviewer's suggestion to show up preliminary results. We have added the number of references we obtained from the search in the electronic databases, however it is not recommended by the journals' guidelines to report preliminary results in protocol papers. (page 5; "The electronic search yielded 4,980 references.").

Minors comments

P4I16-7: Is there a reference to associate to this assertion?

In response to this comment we have changed this assertion. (page 3; "During the past ... perinatal health outcomes.").

P4I21-22: This is per se a limitation for any study. Could the author lay out more clearly the underlining idea?

Non-medical determinants related to a person's social status such as income, educational level and employment status are highly associated. It is therefore important to investigate these different non-medical determinants together in order to determine their association with unfavourable perinatal health outcomes and to provide insight into the way they influence each other. (page 3; "Non-medical determinants ... vulnerable pregnant women.").

P5I54: Although MEDLINE and PubMed are similar, the latter being the public interface, always provide more output as very recent articles are posted. To stick on what you did -or plan to do- it is recommended to opt for one of them.

We have removed PubMed from the text. (page 2; "To retrieve potential ... October 5, 2018." and page 5; "The electronic online ... October 5, 2018.").

P6I24: It is weird the presence of this key word in the equation. I need the author to convince me as it seems contradictory ((mortalit* OR fatal* OR death*) OR surviv*).

Indeed, the search terms may come across contradictory. It is possible that studies report about neonatal survival and/or survival rates instead of perinatal mortality, neonatal mortality, perinatal death, neonatal death and/or mortality rates. We therefore include the search term surviv* in our search strategy in order to avoid missing such studies.

VERSION 2 – REVIEW

REVIEWER	Deborah Randall The University of Sydney, Australia
REVIEW RETURNED	21-Aug-2019

GENERAL COMMENTS	The authors have addressed my previous queries.
---

REVIEWER	Idrissa Beogo Canada
REVIEW RETURNED	26-Aug-2019

GENERAL COMMENTS	Thanks to the author for the effort made to address my comments. I have however noticed that the recommendation related to the 'discussion and or conclusion' was taken into account in the revised version. I strongly hope that the author can formulate a conclusion. Good luck
--